# Viridicatol Isolated from Deep-Sea *Penicillium*
*Griseofulvum* Alleviates Anaphylaxis and Repairs the Intestinal Barrier in Mice by Suppressing Mast Cell Activation

**DOI:** 10.3390/md18100517

**Published:** 2020-10-16

**Authors:** Zhendan Shu, Qingmei Liu, Cuiping Xing, Yafen Zhang, Yu Zhou, Jun Zhang, Hong Liu, Minjie Cao, Xianwen Yang, Guangming Liu

**Affiliations:** 1College of Food and Biological Engineering, Xiamen Key Laboratory of Marine Functional Food, Fujian Provincial Engineering Technology Research Center of Marine Functional Food, Fujian Collaborative Innovation Center for Exploitation and Utilization of Marine Biological Resources, Jimei University, 43 Yindou Road, Xiamen 361021, China; zdanshu@163.com (Z.S.); liuqingmei1229@163.com (Q.L.); zyf1697047824@163.com (Y.Z.); carolzy1205@163.com (Y.Z.); 18899201006@163.com (J.Z.); liuhong@jmu.edu.cn (H.L.); mjcao@jmu.edu.cn (M.C.); 2Key Laboratory of Marine Biogenetic Resources, Third Institute of Oceanography, Ministry of Natural Resources, 184 Daxue Road, Xiamen 361005, China; xingcuiping123@126.com

**Keywords:** food allergy, deep-sea-derived viridicatol, X-ray single crystal, intestinal barrier, mast cell, calcium influx

## Abstract

Viridicatol is a quinoline alkaloid isolated from the deep-sea-derived fungus *Penicillium griseofulvum.* The structure of viridicatol was unambiguously established by X-ray diffraction analysis. In this study, a mouse model of ovalbumin-induced food allergy and the rat basophil leukemia (RBL)-2H3 cell model were established to explore the anti-allergic properties of viridicatol. On the basis of the mouse model, we found viridicatol to alleviate the allergy symptoms; decrease the levels of specific immunoglobulin E, mast cell protease-1, histamine, and tumor necrosis factor-α; and promote the production of interleukin-10 in the serum. The treatment of viridicatol also downregulated the population of B cells and mast cells (MCs), as well as upregulated the population of regulatory T cells in the spleen. Moreover, viridicatol alleviated intestinal villi injury and inhibited the degranulation of intestinal MCs to promote intestinal barrier repair in mice. Furthermore, the accumulation of Ca^2+^ in RBL-2H3 cells was significantly suppressed by viridicatol, which could block the activation of MCs. Taken together, these data indicated that deep-sea viridicatol may represent a novel therapeutic for allergic diseases.

## 1. Introduction

Food allergy is a potentially life-threatening allergic reaction that affects a substantial proportion of the population, being defined as a breakdown of immunological tolerance and clinical symptoms generated in response to ingested food [1]. It is estimated that approximately 8% of children and 5% of adults suffer from food allergies around the world, and the prevalence has also been rising over the past two decades [2]. Food allergy is more common in children than in adults, and it often begins in childhood with the influence of genetic predisposition. 

Eggs are indispensable nutrients in children’s diet pagodas. However, the prevalence of egg allergy in children ranges from 1.2% to 2.0%. The majority allergens within eggs have been found in egg white, with ovalbumin (OVA) being the most abundant form (at 58%, although it is not the major allergen), which is claimed to frequently elicit allergic reactions [3,4]. Thus, the OVA-induced mice allergy model is often used to evaluate the anti-allergic activity of various materials [5,6]. 

Food allergy is characterized by a type I hypersensitivity reaction mediated by immunoglobulinE (IgE), and is often accompanied by adverse reactions, including diarrhea, intestinal injury, and hypothermia [7]. Moreover, it is also accompanied by increased intestinal permeability, which is positively correlated with symptom severity, which persists even in a diet without allergens [8]. Food particles and allergens can cross the epithelial barrier and cause allergic reactions induced by IgE production and mast cell (MC) recruitment [9]. In addition, the development of food allergy involves a variety of immune cells, which play different roles in the process. B lymphocytes are mainly responsible for releasing antibodies (e.g., IgE), which are associated with basophil and MC activation [10]. Cellular immunity is primarily characterized by T lymphocytes, including T helper cells (Th) 1 and 2, and regulatory T cells (Tregs). CD4 + Foxp3 + Tregs can secrete transforming growth factor or interleukin (IL)-10, suppress MC activation, and modulate T and B cells, thereby promoting immunosuppression and relieving anaphylaxis [11,12].

As crucial and rapid immune effector cells, MCs play the key role in IgE-mediated allergic responses [13]. Several reports have indicated that targeting MCs may represent a promising strategy for the treatment of food allergies [14]. MCs are activated when IgE binds with high affinity to the IgE receptor (FcεRI) on the surface of MCs [15]. When re-exposed to specific allergens, MCs were activated with a cascade of tyrosine phosphorylation, and then released a series of allergic mediators [16,17]. During the process of MC activation and degranulation, Ca^2+^ is an essential co-factor that regulates the release of various mediators and granule-plasma membrane fusion [18]. It has been reported that glycyrrhizic acid can perform a function as an “MC stabilizer” and inhibit MC degranulation by suppressing Ca^2+^ influx [19]. Therefore, obtaining a stabilizer that can regulate Ca^2+^ influx may be important for inhibiting MC activation and degranulation.

Novel natural compounds from marine resources have attracted increased attention in recent years due to their potential medicinal value [20]. Due to their unique habitat, deep-sea fungi readily produce various secondary metabolites, which have novel structure and distinctive biological properties. Most fully researched secondary metabolites of deep-sea fungi have mainly been from a limited number of genera, including *Penicillium*, *Aspergillus*, *Fusarium*, and *Cladosporium* [21]. Within the facultative marine fungi, species of *Penicillium* and *Talaromyces* are particularly well known for their ability to produce important bioactive compounds [22]. *Penicillium*, an important genus of the phylum Ascomycota, is widely distributed and has a large impact on human life [23]. Moreover, as one of the most studied fungi, *Penicillium* was considered to be a primary source of drug discovery, which produces a wide range of highly active metabolites, including griseofulvin and the blockbuster drug penicillin [24,25]. After the discovery of penicillin, isolating secondary metabolites of *Penicillium* species has received a remarkable amount of interest from researchers due to the interesting structures and possible pharmaceutical applications of the compounds. It has been shown that *Penicillium* metabolites possess a wide range of biological properties such as antibacterial activities, antioxidant properties, and cytotoxic activities against cancer lines [22]. Thus, as the second most common genus of marine fungi, it is of good importance to investigate the properties and metabolites of *Penicillium* [22]. It has been reported that cyclopiane-type diterpenes from the deep sea-derived fungus *Penicillium* possess antimicrobial activity [26]. However, few systematic studies have investigated the anti-food allergy properties of *Penicillium* secondary metabolites, especially for deep-sea *Penicillium*.

In previous studies, the rat basophil leukemia (RBL)-2H3 cell model was established to explore the anti-allergic potential metabolite from deep-sea fungi including *Penicillium*, *Aspergillus*, *Actinomycete*, and so on [27,28]. The metabolites of deep-sea-derived *Aspergillus* and *Actinomycete* relieved allergy by regulating MC function and accelerating MC apoptosis, respectively. Additionally, nine compounds isolated from deep-sea fungus *Penicillium griseofulvum* were tested for their anti-allergic bioactivity using the RBL-2H3 cell model [29]. A further study on the fungus led to the isolation of another potent anti-allergic compound, viridicatol. As reported, viridicatol effectively inhibited the protein activities and expressions of Matrix matalloproteinases (MMP)-2 and -9 against cancer and tumor [30]. However, the investigation of the biological activity of viridicatol is limited at present. This study was aimed at seeking to further validate and explore the associated mechanism of viridicatol using mouse model, flow cytometry, and an RBL-2H3 cell model. The findings of this study are expected to provide a foundation for the development of new marine anti-allergic drugs.

## 2. Results

### 2.1. Structural Determination of Viridicatol

The chemical structure of viridicatol (Figure 1A) was determined primarily by a detailed analysis of its 1D and 2D nuclear magnetic resonance (NMR) spectra (Appendix A). 

Viridicatol: ^1^H-NMR (CD_3_OD, 400 MHz) δ_H_ 7.33 (1H, m, H-5), 7.32 (1H, m, H-8), 7.31 (1H, d, J = 7.6, H-5′), 7.23 (1H, d, J = 8.2, H-7), 7.11 (1H, m, H-6), 6.86 (1H, dd, J = 9.2, 2.1, H-4′), 6.81 (1H, d, J = 8.1, H-6′), 6.80 (1H, s, H-2′); ^13^C-NMR (CD_3_OD, 100 MHz) δ_C_ 160.5 (s, C-2), 127.3 (s, C-3), 134.4 (s, C-4), 128.0 (d, C-5), 123.8 (d, C-6), 126.3 (d, C-7), 116.5 (d, C-8), 136.2 (s, C-1′), 117.9 (d, C-2′), 158.7 (s, C-3′), 116.0 (d, C-4′), 136.0 (d, C-5′), 122.2 (d, C-6′), 123.0 (s, C-4a), 143.1 (s, C-8a). HRESIMS *m/z* 252.0744 [M − H]^−^.

### 2.2. X-ray Crystallography of Viridicatol

The structure of viridicatol was assigned by the single X-ray (Figure 1B). It was obtained as a colorless needle. The crystal data were recorded with an Xcalibur Eos Gemini single-crystal diffractometer with Cu-Kα radiation (λ = 1.54184 Å). Space group P (−1), a = 6.2014 (2) Å, b = 10.2629 (4) Å, c = 10.7415 (5) Å, *α* = 117.029 (4), *β* = 96.135 (3), *γ* = 100.680 (3), V = 583.98 (5) Å^3^, Z = 2, D_calcd_ = 1.491 g/cm^3^, *μ* = 0.888 mm^−1^, F (000) = 274.0. The final *R* indices were [I > 2σ(I)] *R*_1_ = 0.0524, w*R*_2_ = 0.1410. Crystallographic data of viridicatol were deposited in the Cambridge Crystallographic Data Center (deposition number: 2008745).

### 2.3. Viridicatol Decreased the Release of β-Hexosaminidase and Histamine in RBL-2H3 Cells

In the preliminary screening, we found that viridicatol significantly decreased the release of β-hexosaminidase and histamine in RBL-2H3 cells in a dose-dependent manner (Figure 1C,D). Meanwhile, we found that viridicatol had no measurable cytotoxic effects on RBL-2H3 cells (data not shown), and the IC_50_ value of viridicatol was 6.67 ± 0.6 µg/mL (26.3 µM).

### 2.4. Viridicatol Relieved OVA-Induced Allergic Symptoms in Mice

An OVA-induced mouse model of food allergy was established to explore the anti-allergic activity of viridicatol in vivo (Figure 2A). As expected, the mice in the OVA group exhibited diarrhea, hypothermia, and a high anaphylactic score after five successive oral challenges [31]. Moreover, after daily treatment with viridicatol, the hypothermia (Figure 2B), loose stool rate (Figure 2C), and anaphylactic score (Figure 2D) were significantly attenuated. Compared with the phosphate-buffered saline (PBS) group, the cecum in the OVA group was significantly increased, which was caused by allergic diarrhea. Following treatments with viridicatol, the onset of diarrhea was suppressed (Figure 2E).

### 2.5. Effects of Viridicatol on Immunoglobulins, Anaphylactic Mediators, and Cytokines in the Mouse Serum

Severe allergic reactions are associated with the increased secretion of allergic mediators. In this study, viridicatol significantly decreased the level of OVA-specific IgE at a concentration of 20 mg/kg (Figure 3A). However, no effects of viridicatol on OVA-specific IgG1 and IgG2a were observed (Figure 3B,C). As shown in Figure 3D–F, the oral administration of viridicatol significantly decreased the levels of serum histamine, mast cell protease-1 (mMCP-1) , and tumor necrosis factor-α (TNF-α) in a dose-dependent manner compared to that of the OVA group. As shown in Figure 3G, following an oral administration of viridicatol, the level of interleukin (IL)-10 was increased in the serum as a slight trend toward a dose-dependent manner.

### 2.6. Viridicatol Alleviated Intestinal Injury and Inflammation of the Jejunum

While the jejunum villi were regularly arranged without fracture on the surface of the jejunum villi in the PBS group, they were severely damaged in the OVA group. As expected, the damaged jejunum villi were repaired to different degrees following treatments with viridicatol (Figure 4A). In parallel with the clinical assessment, there were obvious signs of inflammation, including serious lymphocytic infiltration using hematoxylin and eosin (H&E). These symptoms were relieved following treatments with viridicatol (Figure 4B).

### 2.7. Effects of Viridicatol on the Subpopulation of B Cells and Tregs in the Spleen

As previously reported, regulating the populations of immune cells involved in the allergic responses is one of the most effective methods to treat food allergies [6,32]. To seek the target cells of viridicatol in vivo, we isolated the splenic lymphocytes from all groups of mice on day 41, and the lymphocyte populations were detected by flow cytometry. Compared to the PBS group (54.04%), the population of CD19^+^ B cells significantly increased to 64.32% in the OVA group. Moreover, the population of CD19^+^ B cells in the viridicatol group was significantly decreased at a concentration of 20 mg/kg (Figure 5A).

Considering the effects of viridicatol on the level of serum IL-10 (Figure 3G), the population of Tregs in the spleen was further investigated. Compared to the PBS group (19.60%), the population of Tregs was significantly decreased in the OVA group (6.6%). As expected, the population of Tregs was significantly upregulated following treatment with viridicatol (Figure 5B). In summary, viridicatol displayed a weak and lack of a dose-dependent effect on the population of B cells and Tregs, respectively.

### 2.8. Effects of Viridicatol on MC Population and Degranulation

In light of the significant inhibitory effects on histamine and mMCP-1 in the serum, we determined the various populations of FcεRI^+^c-KIT^+^ cells (MCs) by flow cytometry. Compared to the PBS group (0.16%), the population of MCs was significantly increased in the OVA group (1.34%). Following treatments with viridicatol, the population of MCs was significantly decreased compared to the OVA group (Figure 6A). The population of MCs was inhibited after treatments with viridicatol using toluidine blue staining (Figure 6B).

### 2.9. Viridicatol Decreased the Level of Intracellular Calcium

The elevation of Ca^2+^ is a critical process for MC secretory granule translocation [33]. As a calcium ionophore, A23187 is often used to study the status of intracellular Ca^2+^ influx [34]. Moreover, the elevation of intracellular Ca^2+^ stimulated by A23187 was found to be much stronger than that observed in IgE antigen-stimulated cells [16]. In the A32187-induced degranulation model, viridicatol significantly inhibited the release of β-hexosaminidase induced by A23187 in RBL-2H3 cells (Figure 7A). To further study the effect of viridicatol on intracellular Ca^2+^, we used the Ca^2+^ probe to detect the concentration of intracellular Ca^2+^. The concentration of intracellular Ca^2+^ was significantly decreased by viridicatol in a dose-dependent manner compared with the A23187 group, as expected (Figure 7B,C). These results indicated that viridicatol may inhibit MC degranulation by suppressing the influx of intracellular Ca^2+^. 

### 2.10. Relationship between the Chemical Structure and Inhibitory Effects on Cell Degranulation

The activity of compound is closely related to its chemical structure. Two homologues of viridicatol, viridicatin and 3-*O*-methylviridicatin, were also isolated from *Penicillium griseofulvum*. The chemical structures are presented in Figure 8A–C. In light of the effects of viridicatol on OVA-induced food allergy and the RBL-2H3 cell model, we investigated the effects of viridicatol and its homologous on RBL-2H3 cell degranulation to analyze their inhibitory activity. The effect of these compounds on β-hexosaminidase release in antigen and A23187-induced RBL-2H3 cells was measured at a concentration of 10 μg/mL. In the two models, viridicatol displayed an obvious inhibitory effect on β-hexosaminidase release compared to viridicatin (Figure 8D,E), which may be attributed to the missing conjugation of phenolic hydroxyl. Interestingly, 3-*O*-methylviridicatin hardly inhibited RBL-2H3 cell degranulation, which may have been due to the absence of a phenolic hydroxyl, and the hydroxyl on the mother ring was replaced by a methoxy group.

## 3. Discussion

Due to their extreme environments, deep-sea fungi produce secondary metabolites with novel structures and potent bioactivities [21]. Recently, some secondary metabolites from deep-sea-derived fungi have been discovered to possess biological activities; however, they were poorly studied and seldom utilized. We have found that butenolides, polyketides, and penigrisacids have significant anti-allergic activities [27,29]. Unfortunately, few compounds have been evaluated in mouse models to further elucidate the associated mechanisms in vivo. In this study, viridicatol, a type of quinolone alkaloid, was reported as having anti-tumor activities [30]. However, the anti-allergic activity of viridicatol has not been reported. In this study, viridicatol was found to exhibit significant inhibitory effects on RBL-2H3 cell degranulation. In an OVA-induced mouse model, viridicatol significantly alleviated the allergic symptoms of diarrhea and hypothermia.

Food allergy is accompanied by immune disorders, which involves a variety of immune cells and cytokines [35]. In our study, treatments with viridicatol led to a moderate reduction in the population of splenic B cells, and may contribute to the reduction of serum antigen-specific IgE, which is involved in the process of MC and basophil activation [10]. To some extent, viridicatol was able to upregulate the population of Tregs, which is consistent with increased levels of IL-10. Tregs and the associated cytokine, IL-10, was reported to maintain immune balance and regulate MC activation [36]. MCs are the effector cells involved in inflammation and anaphylaxis, playing a crucial role in allergic reactions [13]. Notably, viridicatol decreased the population of MCs in a dose-dependent manner. Meanwhile, viridicatol could significantly decrease the concentration of serum mMCP-1, TNF-α, and histamine, which was related to a reduction of MC numbers and the inhibition of MC degranulation. On the basis of the results of the OVA-induced model of food allergy, viridicatol was demonstrated to have a stronger effect on MCs than B and Treg cells. Thus, viridicatol may relieve food allergy in mice by decreasing the number and inhibiting the degranulation of MCs.

Previous studies have demonstrated that the occurrence of food allergy is related to damage of intestinal barrier function [37]. The intestinal barrier can be divided into mechanical and immune barriers. Moreover, food allergy can influence intestinal barrier function, which can also enhance the speed of food allergy developments [8]. In this study, viridicatol was found to significantly repair the intestinal barrier by reducing the damage of jejuna tissue villi and alleviating the degree of tissue inflammation. MCs residing in the mucosa of the gastrointestinal tract, as the center of the intestinal immune network, are not only limited to the antigen response, but are also involved in the regulation of the intestinal epithelial barrier and transport, changes in mucosal functionality, and production or amplification of signals to other cells [37]. Following treatments with viridicatol, the length of the cecum was significantly reduced. The activation of MCs resulted in diarrhea, which was likely a part of the mucosal defense response [38]. Bischoff et al. demonstrated that histamine plays a multifunctional role in the intestinal barrier [38]. In addition, Jacobs et al. found that MCs released proteases and TNF-α, which resulted in tight junction protein rearrangement and increased the colic epithelium permeability and intestinal inflammation [39]. The level of mMCP-1 in the blood reflected the extent of degranulation by mast cells in the local tissue [40]. Viridicatol reduced the number of MCs in the jejunum tissue, as well as inhibited MC degranulation and the release of mMCP-1, TNF-α, and histamine, which may be important to the repair of the intestinal barrier following exposure to a food allergen.

At the intersection of many classical signaling pathways, Ca^2+^ converts the received extracellular signals into intracellular signals and regulates lymphocyte activation, differentiation, and various transcriptional processes [41]. Ca^2+^ influx is considered to be a necessary condition for triggering immune function and a key factor for MC activation [42,43]. Ca^2+^ promotes MC degranulation, mediator release, and the activation of gene expression to accelerate a subsequent immune response [44]. Viridicatol was found to suppress Ca^2+^ influx, which was closely associated with MC degranulation. Jacobs et al. suggested that the destruction of the intestinal barrier was accompanied by MC degranulation and increased Ca^2+^ [42]. Therefore, the inhibitory effects of viridicatol on MC degranulation and Ca^2+^ influx may be beneficial for intestinal barrier repair and for relief of food allergy symptoms.

Many anti-allergic drugs have some undesirable adverse effects such as dizziness and drowsiness. Thus, the anti-allergic activities of natural sources have attracted our attention due to their low side effects. As reported, terpenoids, flavenoids, and alkaloids have significant anti-allergic activities and different mechanisms of action [29,45,46]. Alkaloids are nitrogen-containing compounds that have many biological activities, including antiviral, anti-inflammation, and anti-allergy properties. Costa pointed that alkaloids may be associated with the inhibition of eosinophil and mast cell activities [46]. Viridicatol is a quinoline alkaloid that may be absorbed more readily due to the small molecular weight (253.26). Moreover, viridicatol was isolated from the deep-sea fungus *Penicillium griseofulvum*, which produces a variety of anti-allergic metabolites [29]. In this study, viridicatol inhibited the degranulation of RBL-2H3 cells and relieved the allergic reactions in mice, which may be related to its structure and thus it is worth further exploring. The bioactivity of these compounds is closely related to their structures. For example, the number and position of hydroxyl groups have been shown to be associated with anti-allergic activities [45]. Moreover, both in antigen-induced and A23187-induced RBL-2H3 cell degranulation models, the inhibitory effect of viridicatol on β-hexosaminidase were stronger than viridicatin, which may due to the presence of an extra phenolic hydroxyl. Interestingly, no significant effect of 3-*O*-methylviridicatin on the release of β-hexosaminidase were observed, which may have resulted from the different chemical structure of 3-*O*-methylviridicatin compared with viridicatol: one lost hydroxyl in ring C and one hydroxyl was replaced by a methoxy in ring B. Phenolic hydroxyl groups have been reported to contribute to the increased efficiency of these compounds [47].

Taken together, deep-sea-derived viridicatol could relieve OVA-induced allergic symptoms. Treatments with viridicatol had significant effects on decreasing the level of anti-OVA-IgE, histamine, mMCP-1, and TNF-α. Moreover, viridicatol significantly upregulated the level of IL-10 in the serum. Viridicatol treatment was also found to downregulate the populations of B cells and MCs in the spleen, as well as upregulate Tregs in the spleen. Additionally, viridicatol repaired the intestinal barrier and alleviated the degree of tissue infiltration. Notably, viridicatol significantly suppressed MC degranulation in the jejunum of mice, which may be attributable to the decreased intracellular flow of Ca^2+^ in MCs. Compared to many alkaloids, viridicatol has a relatively small molecular weight, allowing it to be absorbed more readily, and it is expected to be stabilizing for MCs, which could possess a higher practical value. In conclusion, this research provided some insight into the prevention of food allergy and the application of marine fungi. Marine fungi have a large impact in terms of application in the pharmaceutical industry, since many of their metabolites have entered the clinical pipeline and have the potential of being exploited as novel drugs. The present findings demonstrated that viridicatol have the potential to be applied in the therapy of food allergies. Further systematic safety assessments and optimal dosage determination are necessary for the clinical application of viridicatol.

## 4. Materials and Methods

### 4.1. Reagents an General Experimental Procedures

Fetal bovine serum (FBS) was purchased from Gibco (Grand Island, NY, USA). Roswell Park Memorial Institute (RPMI) 1640, Eagle’s minimum essential medium (EMEM), penicillin–streptomycin solution, and trypsin 0.25% (1×) solution were purchased from HyClone Co. (Logan, UT, USA). Imject alum was obtained from Thermo Fisher Scientific Inc. (Waltham, MA, USA). Evans Blue, anti-dinitrophenyl (DNP)-IgE, ovalbumin (OVA), calcium ionophore A23187, and 4-methyl-umbelliferyl-N-acetyl-β-D-glucosaminide were obtained from Sigma-Aldrich (St Louis, MO, USA). DNP-bovine serum albumin (BSA) was purchased from Biosearch (Petaluma, CA, USA). Goat anti-mouse IgG1 (ab98693) and IgG2a (ab98698) antibodies were purchased from Abcam (Cambridge, United Kingdom). Calcium kit-Fluo 3-AM was purchased from Dojindo Laboratories (Mashiki, Japan). ELISA (enzyme-linked immunosorbent assay) kits for histamine (EHP184), anti-OVA-IgE (500840), and IL-10 (BMS614INST) were obtained from IBL (Hamburg, Germany), Cayman (Ann Arbor, MI, USA), and eBioscience (San Diego, CA, USA), respectively. Antibodies for flow cytometry analysis were obtained from BioLegend (San Diego, CA, USA) and BD (New York, NY, USA). P-phycoerythrin (PE) anti-mouse CD117(c-Kit) (105807), Allophycocyanin (APC) anti-mouse FcεRIα (134316), PE anti-mouse CD19 (115508), Mouse Th/Treg Phenotyping Kit (560767) were also used. All other chemicals were of analytical grade. The HRESIMS spectrum was recorded on a Waters Xevo G2 Q-TOF mass spectrometer (Waters Corporation, Milford, MA, USA). The NMR spectra were recorded on a Bruker 400 MHz spectrometer (Bruker, Fällanden, Switzerland). The single X-ray crystal data were recorded with an XtaLAB AFC12 Kappa single diffractometer using Cu Kα radiation (Rigaku, Japan). Copies of the data can be obtained, free of charge, on application to Cambridge Crystallographic Data Center (CCDC), 12 Union Road, Cambridge CB21EZ, United Kingdom, (fax: +44(0)-1233-336033; email: deposit@ccdc.cam.ac.uk.

### 4.2. Cell Culture

The RBL-2H3 cell line was purchased from the American Type Culture Collection (ATCC, Manassas, VA, USA). Cells were cultured in EMEM supplemented with 10% FBS, 1% penicillin, and 1% streptomycin at 37 °C in an atmosphere containing 5% CO_2_.

### 4.3. Animals

Female BALB/c mice (aged 6–7 weeks) were purchased from Charles River Laboratories (Beijing, China). Six mice were randomly assigned to each experimental group and all mice were maintained in an automatic light/dark cycle (light periods of 12 h) environment and were acclimatized for 1 week before the experiment.

This experiment was performed in strict accordance with the National Institutes of Health (NIH) Guidelines for the Care and Use of Laboratory Animals. All protocols were approved by the Animal Research Ethics Board of Jimei University (Xiamen, China, no. SCXK 2016-0006).

### 4.4. Viridicatol Extraction, Isolation, and Purification

On the basis of a previous method of fermentation, we defatted the crude extract with petroleum ether and CH_2_Cl_2_ by column chromatography over silica gel[29]. The MeOH layer was evaporated under reduced pressure to provide a defatted extract (55.4 g). The extract was then subjected to column chromatography on silica gel using a CH_2_Cl_2_-MeOH gradient (0→100%, 49 mm × 460 mm) to yield six fractions (Fr. 1−Fr.6). Fr. 5 (40.0 g) was separated by column chromatography over ODS (H_2_O-MeOH, 5→80%, 49 mm × 460 mm) to obtain 15 subfractions (sfrs. 5.1–sfrs. 5.15). Sfr. 5.15 was subjected to column chromatography on Sephadex LH-20 (MeOH) to obtain viridicatol (443.0 mg).

### 4.5. Determination of Histamine and β-Hexosaminidase Release in RBL-2H3 Cells

It has previously been reported that the level of β-hexosaminidase and histamine from RBL-2H3 cells activated by IgE–FcεRI complex were detected [45]. The cellular cytotoxicity of viridicatol was measured using MTT (3-(4,5-dimethyl-2-thiazolyl)-2, 5-diphenyl-2-H-tetrazolium bromide) assay.

### 4.6. OVA-Induced Model of Food Allergy

Groups of mice (each group of six) were sensitized with 2 mg alum and 100 µg OVA in 200 µL of phosphate-buffered saline (PBS) intraperitoneally (i.p.) on days 0 and 14. The mice were challenged with 50 mg OVA in 200 µL PBS by gavage every 3 days from days 28 to 40 for a total of 5 times. The mice in the treatment groups were treated with viridicatol by gavage every day starting on day 27. The mice in the PBS group were sensitized with 2 mg alum in 200 µL PBS and challenged with PBS, which served as the control group. The anti-allergic effect of viridicatol was determined on the basis of diarrhea, anaphylaxis, and rectal temperature [31]; mice were placed alone in cages, their feces were counted to calculate the proportion of loose stools in the total stools, and symptoms were scored from 0 to 5.

### 4.7. Measurement of Immunoglobulins, Anaphylactic Mediators, and Cytokines in the Mouse Serum

On day 41, the mouse eyeball sera were collected. OVA-specific IgE, IL-10, and TNF-α were measured by commercial ELISA kits. OVA-specific IgG1 and IgG2a were measured by an indirect ELISA, as previously reported [48]. Serum was obtained from the tail vein blood after 30 min of the final challenge on day 40 to detect the level of histamine and mMCP-1 using a commercial ELISA kit. All sera were stored at −80 °C.

### 4.8. Morphology and Staining Analysis of the Intestinal Tissue

On day 41, the mice were sacrificed, and the cecum tissue was collected for observation. The jejunum tissue was fixed, dehydrated, and coated with gold-palladium, and then the jejunum villi were observed by Phenom Pro Model desktop scanning electron microscopy (SEM, Phenom-world, Eindhoven, The Netherlands) as previously reported [48]. The jejunum tissue was fixed in 4% paraformaldehyde, and Servicebio was entrusted to conduct hematoxylin and eosin (H&E) staining and toluene blue staining for the jejunal tissues.

### 4.9. Splenic Lymphocyte Population Analysis by Flow Cytometry

As previously reported, single-cell suspensions were obtained on day 41 by density gradient centrifugation from the mouse spleen tissue suspension [45]. The cells were labeled with anti-CD3, anti-CD4, anti-CD19, anti-Foxp3, anti-FcεRI, and anti-c-KIT on the surface. Intracellular expression markers were stained with Foxp3-PerCP-Cy5.5 after fixing and permeabilizing as described in the protocol. The cells were gated and demonstrated by flow cytometry(Millipore, Billerica, MA, USA), and data were analyzed with a Guava easyCyte 6-2L system using Guava Soft 3.1.1 software.

### 4.10. Release of β-Hexosaminidase and Ca^2+^ Influx of RBL-2H3 Cells Induced by A23187

To measure the release of β-hexosaminidase induced by A23187, we inoculated RBL-2H3 cells in 48-well plates for 12 h and incubated them with or without viridicatol for 1 h. The cells were then stimulated with 2.5 µM A23187 for 15 min, after which β-hexosaminidase release was measured. The concentration of Ca^2+^ was measured using a Calcium kit-Fluo 3 according to the manufacturer’s instructions. RBL-2H3 cells (5 × 10^4^ in 100 μL) were seeded into a black 96-well plate and cultured for 16 h. The cells were then washed and incubated with Fluo-3 AM (10 µM) for 1 h. Next, the cells were washed and treated with viridicatol (2.5, 5, 10 µg/mL) in HBSS for 1 h. After we stimulated it with A23187 (2.5 µM), the fluorescent intensity was immediately monitored using the microplate reader at an excitation wavelength of 490 nm and an emission wavelength of 530 nm every 30 s. [Ca^2+^]i was calculated as follows: [Ca^2+^]i (nm) = Kd [(F − Fmin)/(Fmax − F)], Fmin: the background fluorescence of 5 mM Ethylenebis(oxyethylenenitrilo) tetraacetic acid (EGTA), Fmax: the maximum fluorescence with of 0.1% Triton X-100, Kd: the dissociation constant of Ca^2+^ and Fluo-3 (450 nM). Moreover, the cells were photographed with a fluorescence microscope (Echo, San Diego, CA, USA) to observe the variation of intracellular Ca^2+^.

### 4.11. Statistical Analysis

All experiments were repeated at least 3 times, and the data were presented as the mean ± standard deviation (SD). Statistical differences were analyzed using a one-way analysis of variance (ANOVA). A threshold *p*-value of 0.05 or 0.01 indicated a significant difference.

## Figures and Tables

**Figure 1 marinedrugs-18-00517-f001:**
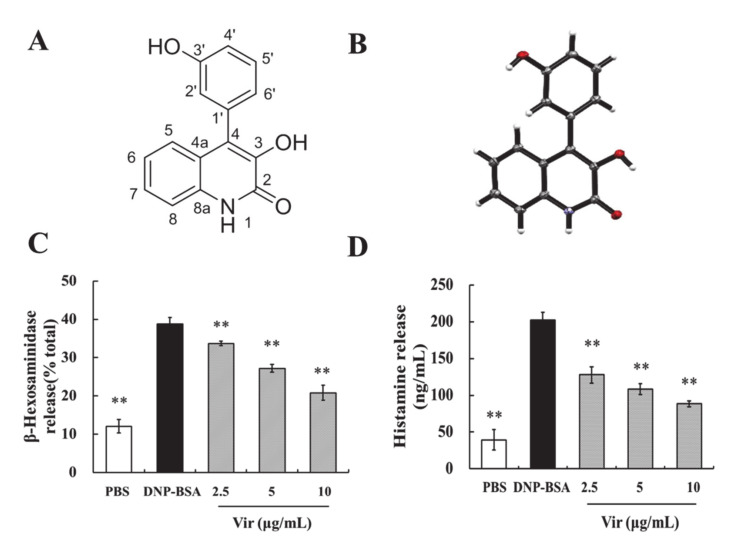
The X-ray crystallography and degranulation efficiency of viridicatol. (**A**) Chemical structure of viridicatol. (**B**) The X-ray crystallography of viridicatol. (**C**) Effects of viridicatol on degranulation. Cells were incubated with 200 ng/mL of dinitrophenyl (DNP)-specific immunoglobulinE (IgE) for 16 h in 48-well plates. The medium was replaced by Tyrode’s buffer containing the indicated concentrations of viridicatol (2.5, 5, and 10 µg/mL) followed by stimulation with 500 ng/mL DNP-bovine serum albumin (BSA) for 1 h. The release of β-hexosaminidase was measured. (**D**) Effects of viridicatol on histamine release. The cells were sensitized and treated as described in (C), except for stimulation with DNP-BSA for 15 min, and the level of histamine was measured using an ELISA kit. ** *p* < 0.01 compared with the DNP-BSA group. The data represent the mean ± SD of three repeated experiments. Vir: viridicatol.

**Figure 2 marinedrugs-18-00517-f002:**
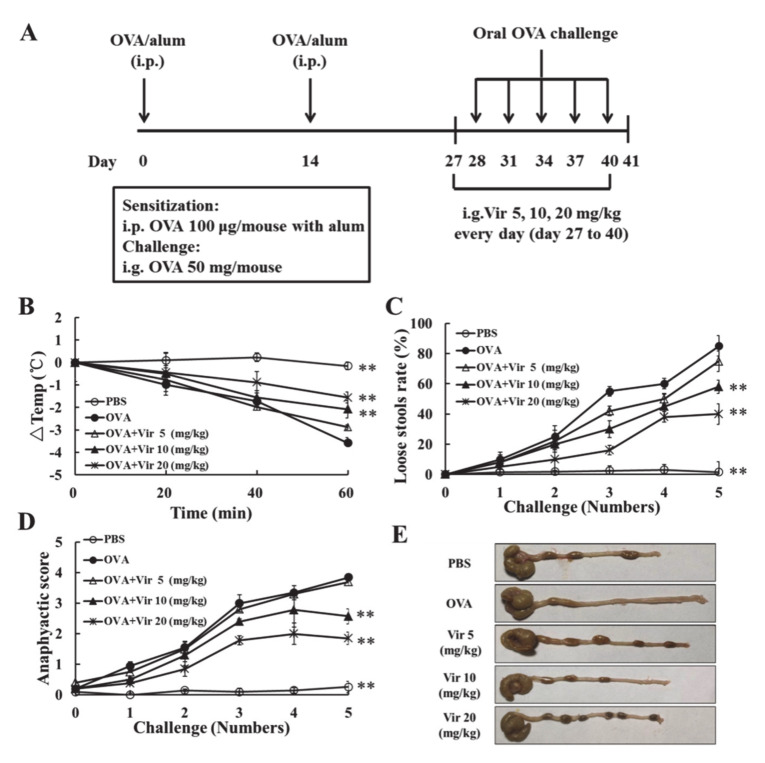
Food allergy model and effects of viridicatol on the allergic symptoms. (**A**) Food allergy model. Mice were sensitized with 100 µg ovalbumin (OVA) and 2 mg alum in 200 µL of phosphate-buffered saline (PBS) by intraperitoneal injection, orally challenged with 50 mg OVA in 200 µL PBS, and the viridicatol concentrates (5, 10, 20 mg/kg) were orally administered. (**B**) The rectal temperature was measured 1 h after the fifth OVA challenge. (**C**) The rate of loose stools was measured 1 h after each OVA challenge. (**D**) The anaphylactic score was measured 1 h after each OVA challenge. (**E**) The representative large macroscopic view of the intestine in each group of mice. ** *p* < 0.01 compared with the OVA group. The data represent the mean ± SD. Vir: viridicatol.

**Figure 3 marinedrugs-18-00517-f003:**
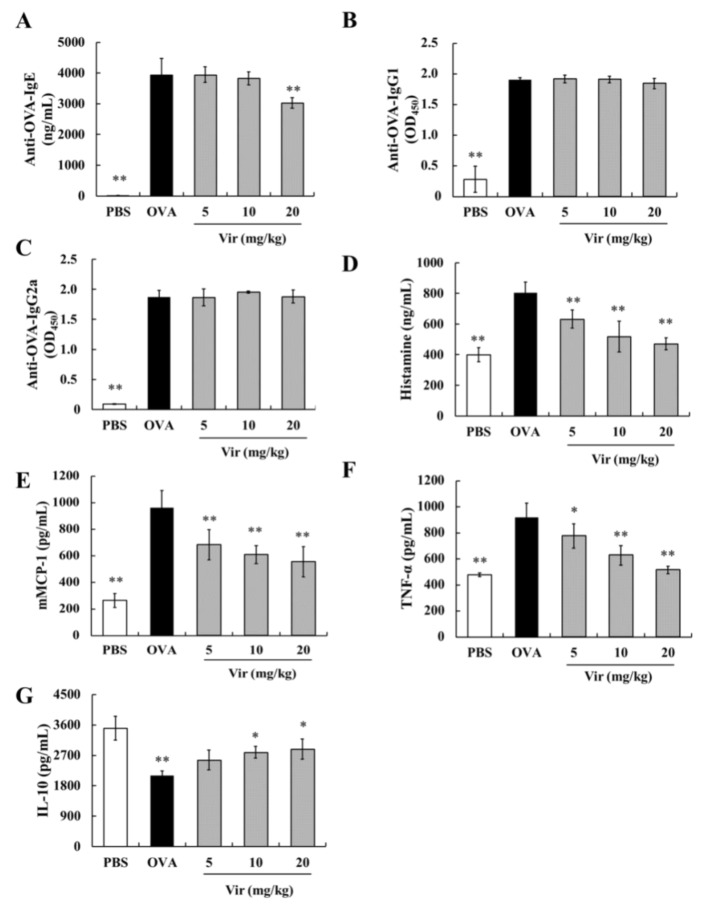
The levels of immunoglobulins, anaphylactic mediators, and cytokines in serum (*n* = 6). (**A**) The level of OVA-specific IgE. (**B**) The level of anti-OVA IgG1. (**C**) The level of anti-OVA IgG2a. (**D**) The level of histamine. (**E**) The level of mMCP-1. (**F**) The level of TNF-α. (**G**) The level of IL-10. In (A,B,C,F,G), serum was from retro-orbital blood collection on day 41. In (D,E), the serum from the mice tail veins was collected within 1 h after the last challenge. * *p* < 0.05, ** *p* < 0.01 compared with the OVA group. The data represent the mean ± SD. Vir: viridicatol.

**Figure 4 marinedrugs-18-00517-f004:**
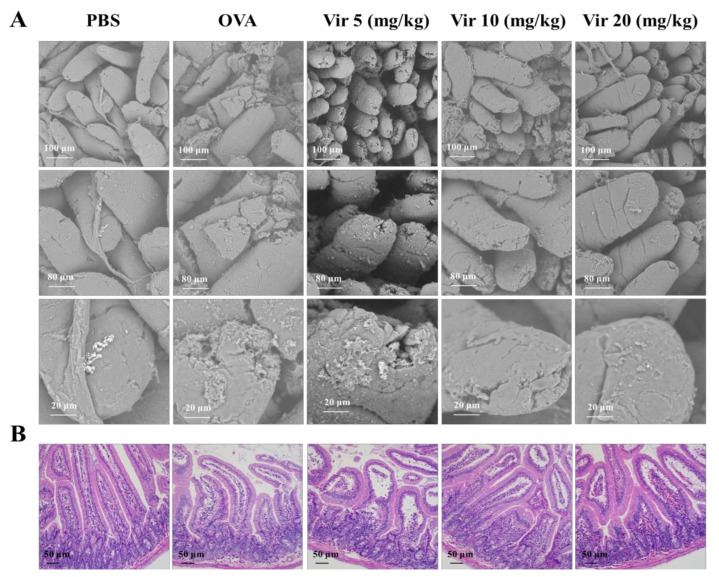
Effects of viridicatol on jejunum tissue injury. (**A**) Injury of the jejunum villi. The scanning electron microscopy (SEM) results shown at 500×, 1000×, and 3000× magnification. (**B**) Representative hematoxylin and eosin (H&E)-stained jejunum sections (magnification: 200×). Vir: viridicatol.

**Figure 5 marinedrugs-18-00517-f005:**
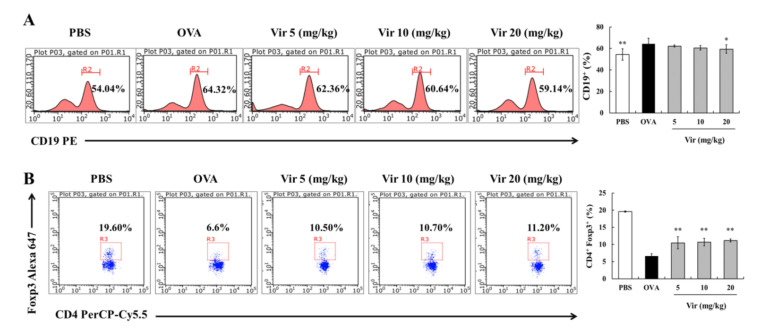
Effects of viridicatol on the population of B cells and regulatory T cells (Tregs) in the spleen. (**A**) Histograms of splenic B cells. Splenic lymphocytes were labeled with anti-CD3 and anti-CD4. (**B**) Scatter diagrams of regulatory T cells (Tregs). Splenic lymphocytes were labeled with anti-CD4 and anti-Foxp3. Spleens were isolated from each group of mice 24 h after the last challenge, labeled with various antibodies, and underwent Fluorescence Activating Cell Sorter (FACS) analysis by flow cytometry. * *p* < 0.05, ** *p* < 0.01 compared with the OVA group. The data represent the mean ± SD. Vir: viridicatol.

**Figure 6 marinedrugs-18-00517-f006:**
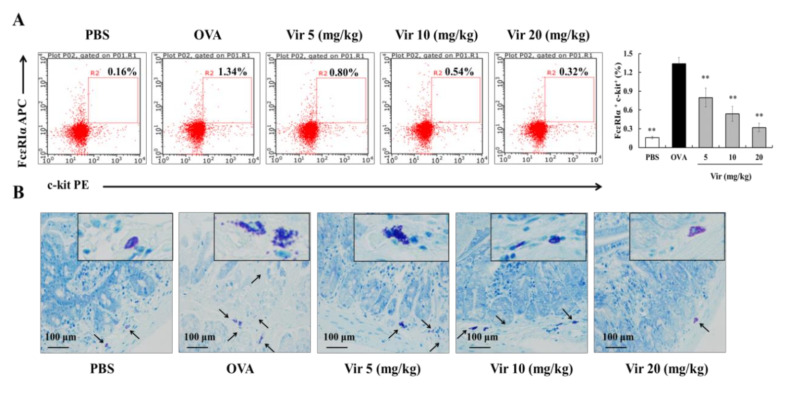
Effects of viridicatol on the mast cell (MC) population and degranulation. (**A**) Scatter diagrams of MCs and FACS analysis by flow cytometry. Splenic lymphocytes were labeled with anti-c-kit and anti-FcεR Iα. (**B**) Representative toluidine blue-stained jejunum sections (magnification: 400×). Jejunum tissues were isolated from each group of mice 24 h after the last challenge and fixed in paraformaldehyde. ** *p* < 0.01 compared with the OVA group. The data represent the mean ± SD. Vir: viridicatol.

**Figure 7 marinedrugs-18-00517-f007:**
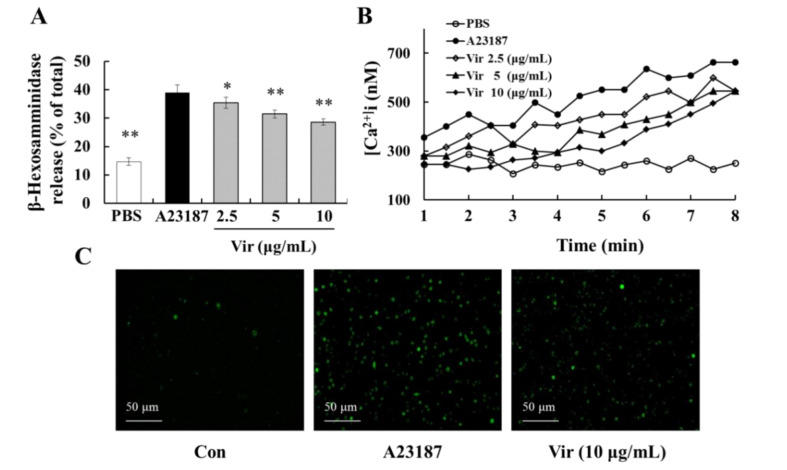
Effects of viridicatol on the degranulation of RBL-2H3 cells induced by A23187 and the concentration of intracellular calcium. (**A**) The release of β-hexosaminidase reduced by the calcium ionophore A23187. RBL-2H3 cells were inoculated in 48-well plates for 12 h andincubated with viridicatol in the presence of 2.5, 5, and 10 μg/mL for 1 h. The cells were then stimulated with 2.5 μM A23187 for 15 min. The release of β-hexosaminidase was measured. (**B**) Effect of Vir on [Ca^2+^]i. (**C**) The accumulation of intracellular calcium (magnification: 200×). * *p* < 0.05, ** *p* < 0.01 compared with the A23187 group. The data represent the mean ± SD of three repeated experiments. Vir: viridicatol.

**Figure 8 marinedrugs-18-00517-f008:**
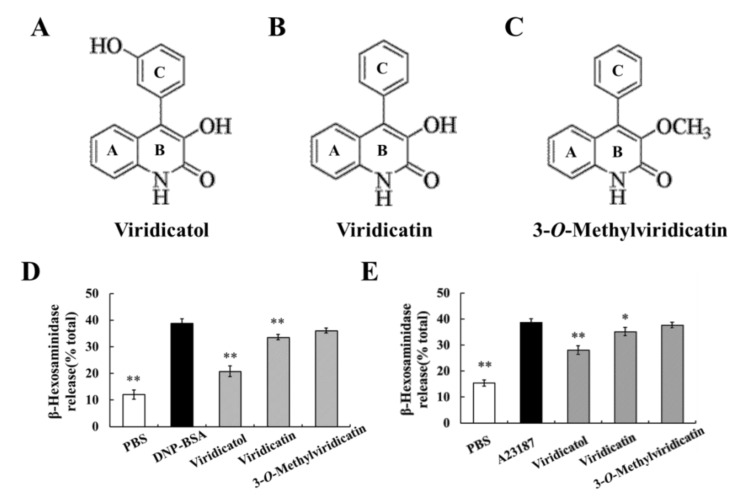
Viridicatol and its homologs for the chemical, structural, and the inhibitory effect on RBL-2H3 cells. (**A**) The chemical structure of viridicatol. (**B**) The chemical structure of viridicatin. (**C**) The chemical structure of 3-*O*-methylviridicatin. (**D**) Effects of viridicatol and its homologs on IgE-induced degranulation. Same as Figure 1C, except that viridicatol was replaced by its homologs at 10 μg/mL. (**E**) Effects of viridicatol and its homologs on A23187-induced degranulation. Same as Figure 7B, except that viridicatol was replaced by its homologs at 10 μg/mL. * *p* < 0.05, ** *p* < 0.01 compared with the DNP-BSA or A23187 groups. The data represent the mean ± SD of three repeated experiments. Vir: viridicatol.

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
