# Peer review of "Viridicatol Isolated from Deep-Sea Penicillium Griseofulvum Alleviates Anaphylaxis and Repairs the Intestinal Barrier in Mice by Suppressing Mast Cell Activation"

_marinedrugs, 2020, doi:10.3390/md18100517_

Round 1

Reviewer 1 Report

This manuscript focuses on a molecule, viridicatol isolated from Penicillium griseofulvum and its potential therapeutic properties for food allergy. Firstly, the authors focus on isolation and structural characterization of the molecule. Next, they have shown that viridicatol is able to suppress IgE cross-linking in an RBL cell line model for IgE reactivity. Using a mouse model it was shown that viridicatol is able to alleviate hypothermia, anaphylactic score, diarhhoea in a dose dependent manner. Further analysis of the mouse model showed that viridicatol treatment led to suppressive effects on allergen-specific IgE, histamine, mMCP and TNFalpha and increasing effect on !L-10. Using histology, FACS analysis and Calcium analysis, it was shown that mast cells were suppressed on viridicatol treatment as well as calcium levels. Overall, the authors have presented an interesting case of a single molecule with anti-allergic effects in a mouse model with good data to further investigate the potential as a therapeutic. Following are my comments and concerns with the manuscript.

Major comments:

  • It is not clear, why and how the authors chose to investigate viridicatol from all the other metabolites found in this penicillium species. There are no mentions on previous literature showing the potential to treat immune and other health diseases for viridicatol. The authors should add more information to show the rationale behind choosing this molecule, and cite previous studies on viridicatol. Mainly in introduction but also in discussion.
  • The introduction section is heavily focused on the details of food allergy and very little on the penicillium, or viridicatol or other similar studies conducted previously. I would suggest elaborating this part more since the most of the work is focused on this molecule.
  • Animal model – was there specific reason to choose this particular immunization/challenge regimen? Why not challenge with allergen, and then feed them the viridicatol to analyse the effects.
  • The discussion is well explained and elaborated, however missing a few key topics A) a brief description and comparison to other purified molecules from other natural sources which also have been investigated for anti-allergic effects. B) The rationale and reasoning underlying the selection of viridicatol for testing its potential therapeutic effects, and C) what are the potential caveats in this study, particularly in the use of viridicatol.

Minor comments:

  • Line 54 – please use a more recent article for reference 9
  • Lines 61-63 – please check and correct grammatical errors
  • Line 69 – where is glycrrhizic acid from? Please elaborate
  • Section 2.1 – doesn’t provide much details of the experimental setup ( as a non expert of NMR)
  • Line 114 – how long were the Ige RBL-2H3 cells incubated with viridicatol?
  • Were the isolated fraction tested for endotoxins before animal work?
  • Figure 2- please use coloured symbols or bigger fonts in fig 2B, C and D. It is very difficult to distinguish the different treatments in its current form.
  • Figure 3 – please state the number of mice (n) used to demonstrate the changes in different Igs and cytokines. What test was used for the statistical analysis?
  • Figure 4 – states that the H&E slides are at 200x, can you please confirm? The image doesn’t look like 200 X image.
  • Section 2.10 – It is not clear why the authors chose to compare the other molecules to viridicatol at this stage? Please state the importance in discussion. Also, the outcomes of viridacatin has not been mentioned.
  • Line 298 – may ‘have’ resulted
  • Some methods sections are too little explained. A bit more elaboration is required to fully understand the experimental setup conditions, and commercial kits that were used.
  • Line 41- Period is missing in after the statement
  • Line 43– I do not necessarily agree with the statement as I do not know what curative treatment authors are referring to in that statement.
  • Line 61- Please correct the following “When re-exposured” to “When re-exposed”.
  • Line 72- Please correct the following “attribute” to “attributing”.
  • Line 79- Please correct the following “possesses” to “possess”.
  • Line 84/85 – Please check whether the statement should be in past tense?
  • Line 142- Please correct the grammar in the statement “However, no effects of viridicatol on OVA-specific IgG1 and IgG2a were not observed”
  • Line 146- It looks like that the increase in the IL-10 is dose dependent as slight trend can be observed, whereas authors have stated it is not. Is there any reason?
  • Figure 3 caption/ Line 151- Please change the statement stating that blood was collected from the eyeball, rather state it as retro-orbital blood collection.
  • Line 155- The statement is confusing as I do not understand what does author mean by ruptures?
  • Line 156- Please correct the following “broken” to “damaged”.
  • Please re-check the statistics in figure 5A in 20mg/kg group.
  • Line 191/192- Please correct the statement as toluidine blue staining was just used to observe the MC population.

Author Response

Response to Reviewer 1 Comments

Point 1: It is not clear, why and how the authors chose to investigate viridicatol from all the other metabolites found in this penicillium species. There are no mentions on previous literature showing the potential to treat immune and other health diseases for viridicatol. The authors should add more information to show the rationale behind choosing this molecule, and cite previous studies on viridicatol. Mainly in introduction but also in discussion.

Response 1: Thank you very much for careful reviewing of our manuscript and we are grateful for your advice. Actually, the present study is a subsequent investigation of previous work. In previous study, the rat basophil leukemia (RBL)-2H3 cell model was established to explore the anti-allergic potential metabolite from deep-sea fungi including Penicillium, Aspergillus, Actinomycete and so on [1, 2]. And the metabolites of deep-sea-derived Aspergillus and Actinomycete relieved allergy by regulating MCs function and accelerating MCs apoptosis respectively. Additionally, nine compounds isolated from deep-sea fungus Penicillium griseofulvum were tested for their anti-allergic bioactivity using the RBL-2H3 cell model [3]. A further study on the fungus led to the isolation of another potent anti-allergic compound, viridicatol. As reported, viridicatol effectively inhibited the protein activities and expressions of MMP-2 and -9 to against cancer and tumor [4]. However, the investigation of biological activity of viridicatol is limited at present. This study was aimed at seeking to further validate and explore the associated mechanism of viridicatol using mouse model, flow cytometry, and RBL-2H3 cell model. The findings of this study are expected to provide a foundation for the development of new marine anti-allergic drugs. Thanks for your suggestion, we have supplemented this part in line 90-101 in the revised manuscript. Point 2: The introduction section is heavily focused on the details of food allergy and very little on the penicillium, or viridicatol or other similar studies conducted previously. I would suggest elaborating this part more since the most of the work is focused on this molecule. Response 2: Thank you very much for your advice. Based on the introduction of penicillium, the relevant descriptions has been modified as “Within the facultative marine fungi, species of Penicillium and Talaromyces are particularly known for their ability to produce important bioactive compounds [5]. Penicillium,an important genus of the phylum Ascomycota,is widely distributed and has a large impact on human life [6]. Meanwhile, as one of the most studied fungi, Penicillium was considered to be a primary source of drug discovery, which produces a wide range of highly active metabolites, including griseofulvin and the blockbuster drugs penicillin. After the discovery of penicillin, isolating secondary metabolites of Penicillium species, has received a remarkable interest of researchers due to the interesting structures and possible pharmaceutical applications of the compounds. It has been shown that Penicillium metabolites possess a wide range of biological properties, such as antibacterial activities, antioxidant properties, and cytotoxic activities against cancer lines [5]. Thus, as the second most common genus of marine fungi, it is of good importance to investigate the properties and metabolites of Penicillium [5]”. It has been added in line 75-86 in the revised manuscript.Besides, the introduction of viridicatol and other similar previous studies have been changed as “In previous study, the rat basophil leukemia (RBL)-2H3 cell model was established to explore the anti-allergic potential metabolite from deep-sea fungi including Penicillium, Aspergillus, Actinomycete and so on [1, 2]. And the metabolites of deep-sea-derived Aspergillus and Actinomycete relieved allergy by regulating MCs function and accelerating MCs apoptosis respectively. Additionally, nine compounds isolated from deep-sea fungus Penicillium griseofulvum were tested for their anti-allergic bioactivity using the RBL-2H3 cell model [3]. A further study on the fungus led to the isolation of another potent anti-allergic compound, viridicatol. As reported, viridicatol effectively inhibited the protein activities and expressions of MMP-2 and -9 to against cancer and tumor [4]. However, the investigation of biological activity of viridicatol is limited at present”. It has been added in line 90-99 in the revised manuscript.

Point 3: Animal model – was there specific reason to choose this particular immunization/challenge regimen? Why not challenge with allergen, and then feed them the viridicatol to analyse the effects.

Response 3: Thank you very much for careful reviewing of our manuscript As reported, this classic mice food allergy model could effectively constructed by this immunization regimen [7, 8]. It has been widely used in the exploration of anti-food allergy active substance. In our study, mouse model was carried out with reference to this food allergy model. Mice were sensitized by intraperitoneal OVA injection on day 0 and day 14 to product OVA-specific IgE, which is not a rapid process. When the organ exposed to the specific allergen again, rapidly binding between specific IgE and allergen causes a range of allergic reaction, which is recognized as the effect period. OVA were administered to mice every three days starting on day 28 to motivate mice food allergies and establish the food allergy model. During this period, simulate the process of normal drug administration [9], mice were treated with viridicatol to explore the anti-food allergic activity of viridicatol.

Point 4: The discussion is well explained and elaborated, however missing a few key topics A) a brief description and comparison to other purified molecules from other natural sources which also have been investigated for anti-allergic effects. B) The rationale and reasoning underlying the selection of viridicatol for testing its potential therapeutic effects, and C) what are the potential caveats in this study, particularly in the use of viridicatol.

Response 4: Thank you very much for careful reviewing of our manuscript. According to your suggestion, we have supplemented these key topics in appropriate part and discussed around them:

A:The descriptions have been adjusted as “Many anti-allergic drugs have some undesirable adverse effects such as dizziness and drowsiness. Thus, the anti-allergic activities of natural sources have attracted our attention due to its low side effects. As reported, terpenoids, flavenoids, and alkaloids have significant anti-allergic activities and different mechanisms of action [3, 10, 11] ” in the line 314-317 in the revised manuscript.

B: The descriptions have been modified as “Viridicatol is a quinoline alkaloid which may be absorbed more readily due to the small molecular weight (253.26). And viridicatol was isolated from deep-sea fungus Penicillium griseofulvum, which produced a variety of anti-allergic metabolites [3]” in the line 320-322 in the revised manuscript.

C: The descriptions have been adjusted and added as “Marine fungi have the large impact on the application of pharmaceutical industry, since many metabolites from them have entered the clinical pipeline and in view of being exploited as novel drugs. The present findings demonstrated that viridicatol have the potential to be applied in the therapy of food allergy. Further systematic safety assessments, optimal dosage determination are necessary for the clinical application of viridicatol” in the line 344-349 in the revised manuscript.

Point 5: Line 54 – please use a more recent article for reference 9

Response 5: Thanks for your careful reviewing of our manuscript. The replaced citation of this paragraph is “Takhar, P.; Smurthwaite, L.; Coker, H. A.; Fear, D. J.; Banfield, G. K.; Carr, V. A.; Durham, S. R.; Gould, H. J., Allergen drives class switching to IgE in the nasal mucosa in allergic rhinitis. Journal of Immunology 2005, 174, 5024-5032 .” [12]

Point 6: Lines 61-63 – please check and correct grammatical errors

Response 6: Thanks for your careful reviewing of our manuscript. The descriptions have been corrected to “When re-exposed to specific allergens, MCs were activated with a cascade of tyrosine phosphorylation, and then released series of allergic mediators” in line 63-64 in the revised manuscript.

Point 7: Line 69 – where is glycrrhizic acid from? Please elaborate

Response 7: Thanks for your careful reviewing of our manuscript. As reported, the glycyrrhizic acid (G111375) was obtained from Aladdin biochemical technology (Shanghai, China) [9].

Point 8: Section 2.1 – doesn’t provide much details of the experimental setup ( as a non expert of NMR)

Response 8: Thanks for your careful reviewing of our manuscript. A new version of 1D and 2D NMR spectra were provided in the revised Supporting Information, so that reviewers could easily find details of the experimental setup.

Point 9: Line 114 – how long were the Ige RBL-2H3 cells incubated with viridicatol?

Response 9: Thanks for your careful reviewing of our manuscript. In our study, RBL-2H3 cells were incubated with 200 ng/mL of DNP-specific IgE for 16 h in 48-well plates. The description has been added to line 133 in the revised manuscript.

Point 10: Were the isolated fraction tested for endotoxins before animal work?

Response 10: Thanks for your careful reviewing of our manuscript.

The isolated fraction contains less than 0.06 EU/mg of endotoxin by Limulus reagent kit (G010060) (Xiamen Bioendo Technology Co., Ltd), which could not affect the present result.

Point 11: Figure 2- please use coloured symbols or bigger fonts in fig 2B, C and D. It is very difficult to distinguish the different treatments in its current form.

Response 11: Thank you very much for careful reviewing of our manuscript and we are grateful for your advice. The bigger fonts have been used in fig 2B, C and D in the revised manuscript.

Point 12: Figure 3 – please state the number of mice (n) used to demonstrate the changes in different Igs and cytokines. What test was used for the statistical analysis?

Response 12: Thank you very much for careful reviewing of our manuscript. The number of mice was 6 in each group, and the data were presented as the mean ± standard deviation (SD). Statistical differences were analyzed using a one-way analysis of variance (ANOVA). A threshold P value of 0.05 or 0.01 indicated a significant difference. The descriptions have been added to line 168 and line 168 and 172-173 in the revised manuscript.

Point 13: Figure 4 – states that the H&E slides are at 200x, can you please confirm? The image doesn’t look like 200 X image.

Response 13: Thank you very much for careful reviewing of our manuscript. After confirming, the H&E slides are at 200×. Servicebio company was entrusted to conduct hematoxylin-eosin (H&E) staining for the jejunal tissues using Eclipse ci (lenses: 20×, eyepiece: 10×).

Point 14: Section 2.10 – It is not clear why the authors chose to compare the other molecules to viridicatol at this stage? Please state the importance in discussion. Also, the outcomes of viridacatin has not been mentioned.

.

Response 14: Thank you very much for careful reviewing of our manuscript. The anti-allergic bioactivities, outcomes, and mechanism of viridicatol were explored and elaborated in 2.1-2.9. However, the anti-allergic activities of viridicatin and 3-O-methylviridicatin which are the homologues of viridicatol, are weaker than viridicatol. The activity of compound is closely related to its chemical structure. Thus the relationship between structures and activity were discussed in section 2.10. The relevant description was mentioned in 324-333 in the discussion section.

Point 15: Line 298 – may ‘have’ resulted.

Response 15: Thank you very much for careful reviewing of our manuscript. The expression of “may resulted” has been corrected to “may have resulted” in line 330 in the revised manuscript.

Point 16: Some methods sections are too little explained. A bit more elaboration is required to fully understand the experimental setup conditions, and commercial kits that were used.

Response 16: Thank you very much for careful reviewing of our manuscript and we are grateful for your advice. The material and methods for the NMR, MS, and X-ray crystallography were added and mentioned in the revised section 2.1. and 2.2., respectively. The catalog numbers and origins of commercial kits and antibody have been added in the revised section 4.1.

Point 17: Line 41- Period is missing in after the statement

Response 17: Thank you very much for careful reviewing of our manuscript. The period has been added to line 43 in the revised manuscript.

Point 18: Line 43– I do not necessarily agree with the statement as I do not know what curative treatment authors are referring to in that statement.

Response 18: Thank you very much for careful reviewing of our manuscript. The descriptions have been added as “Many anti-allergic drugs have some undesirable adverse effects, such as dizziness and drowsiness” in line 314-315 in the revised manuscript.

Point 19: Line 61- Please correct the following “When re-exposured” to “When re-exposed”.

Response 19: Thank you very much for careful reviewing of our manuscript and we are grateful for your advice. The expression of “When re-exposured” has been corrected to “When re-exposed” in line 63 in the revised manuscript.

Point 20: Line 72- Please correct the following “attribute” to “attributing”.

Response 20: Thank you very much for careful reviewing of our manuscript. The expression of “attribute” has been corrected to “attributin” in line 71 in the revised manuscript.

Point 21: Line 79- Please correct the following “possesses” to “possess”.

Response 21: Thank you very much for careful reviewing of our manuscript. The expression of “possesses” has been corrected to “possess” in line 87 in the revised manuscript.

Point 22: Line 84/85 – Please check whether the statement should be in past tense?

Response 22: Thank you very much for careful reviewing of our manuscript and we are grateful for your advice. The statement has been corrected and were described as “The findings of this study is expected to provide a foundation for the development of new marine anti-allergic drugs” in line 100-101 in the revised manuscript.

Point 23: Line 142- Please correct the grammar in the statement “However, no effects of viridicatol on OVA-specific IgG1 and IgG2a were not observed”

Response 23: Thank you very much for careful reviewing of our manuscript. The description has been corrected to “However, no effects of viridicatol on OVA-specific IgG1 and IgG2a were observed” in line 161-162.

Point 24: Line 146- It looks like that the increase in the IL-10 is dose dependent as slight trend can be observed, whereas authors have stated it is not. Is there any reason?

Response 24: Thank you very much for careful reviewing of our manuscript. The increase in the IL-10 is dose dependent with slight trend, the related description has been modified in line 165-166 in the revised manuscript.

Point 25: Figure 3 caption/ Line 151- Please change the statement stating that blood was collected from the eyeball, rather state it as retro-orbital blood collection.

Response 25: Thank you very much for careful reviewing of our manuscript and we are grateful for your advice. The description has been corrected to “serum was from retro-orbital blood collection on day 41” in line 171.

Point 26: Line 155- The statement is confusing as I do not understand what does author mean by ruptures?

Response 26: Thank you very much for careful reviewing of our manuscript. The expression of “ruptures” has been changed to “fracture” in line 175 in the revised manuscript.

Point 27: Line 156- Please correct the following “broken” to “damaged”.

Response 27: Thank you very much for careful reviewing of our manuscript and we are grateful for your advice. The expression of “damaged” has been corrected to “broken” in line 176 in the revised manuscript.

Point 28: Please re-check the statistics in figure 5A in 20mg/kg group.

Response 28: Thank you very much for careful reviewing of our manuscript. The statistics were re-checked, statistical differences were analyzed using a one-way analysis of variance (ANOVA), and the P value was less than 0.05 in 20mg/kg group.

Point 29: Line 191/192- Please correct the statement as toluidine blue staining was just used to observe the MC population.

Response 29: Thank you very much for careful reviewing of our manuscript and we are grateful for your advice. The expression of “The degranulation of MCs” has been corrected to “The population of MCs” in line 210 in the revised manuscript.

References:

  1. Gao, Y. Y.; Liu, Q. M.; Liu, B.; Xie, C. L.; Cao, M. J.; Yang, X. W.; Liu, G. M., Inhibitory Activities of Compounds from the Marine Actinomycete Williamsia sp. MCCC 1A11233 Variant on IgE-Mediated Mast Cells and Passive Cutaneous Anaphylaxis. J. Agric. Food Chem. 2017, 65, 10749-10756.
  2. Liu, Q. M.; Xie, C. L.; Gao, Y. Y.; Liu, B.; Lin, W. X.; Liu, H.; Cao, M. J.; Su, W. J.; Yang, X. W.; Liu, G. M., Deep-Sea-Derived Butyrolactone I Suppresses Ovalbumin-Induced Anaphylaxis by Regulating Mast Cell Function in a Murine Model. J. Agric. Food Chem. 2018, 66, 5581-5592.
  3. Xing, C. P.; Xie, C. L.; Xia, J. M.; Liu, Q. M.; Lin, W. X.; Ye, D. Z.; Liu, G. M.; Yang, X. W., Penigrisacids A-D, Four New Sesquiterpenes from the Deep-Sea-Derived Penicillium griseofulvum. Mar. Drugs 2019, 17.
  4. Liang, P.; Zhang, Y. Y.; Yang, P.; Grond, S.; Zhang, Y.; Qian, Z. J., Viridicatol and viridicatin isolated from a shark-gill-derived fungus Penicillium polonicum AP2T1 as MMP-2 and MMP-9 inhibitors in HT1080 cells by MAPKs signaling pathway and docking studies. Med. Chem. Res. 2019, 28, 1039-1048.
  5. Nicoletti, R.; Trincone, A., Bioactive Compounds Produced by Strains of Penicillium and Talaromyces of Marine Origin. Mar. Drugs 2016, 14.
  6. Zhang, P.; Wei, Q.; Yuan, X. L.; Xu, K., Newly reported alkaloids produced by marine-derived Penicillium species (covering 2014-2018). Bioorg. Chem. 2020, 99.
  7. Castillo-Courtade, L.; Han, S.; Lee, S.; Mian, F. M.; Buck, R.; Forsythe, P., Attenuation of food allergy symptoms following treatment with human milk oligosaccharides in a mouse model. Allergy 2015, 70, 1091-1102.
  8. Kim, J. H.; Jeun, E. J.; Hong, C. P.; Kim, S. H.; Jang, M. S.; Lee, E. J.; Moon, S. J.; Yun, C. H.; Im, S. H.; Jeong, S. G.; Park, B. Y.; Kim, K. T.; Seoh, J. Y.; Kim, Y. K.; Oh, S. J.; Ham, J. S.; Yang, B. G.; Jang, M. H., Extracellular vesicle-derived protein from Bifidobacterium longum alleviates food allergy through mast cell suppression. J. Allergy Clin. Immunol. 2016, 137, 507-516.
  9. Han, S. W.; Sun, L.; He, F.; Che, H. L., Anti-allergic activity of glycyrrhizic acid on IgE-mediated allergic reaction by regulation of allergy-related immune cells. Sci. Rep. 2017, 7.
  10. Zhang, Y. F.; Liu, Q. M.; Liu, B.; Shu, Z. D.; Han, J.; Liu, H.; Cao, M. J.; Yang, X. W.; Gu, W.; Liu, G. M., Dihydromyricetin inhibited ovalbumin-induced mice allergic responses by suppressing the activation of mast cells. Food Funct. 2019, 10, 7131-7141.
  11. Costa, H. F.; Leite, F. C.; Alves, A. F.; Barbosa-Filho, J. M.; dos Santos, C. R.; Piuvezam, M. R., Managing murine food allergy with Cissampelos sympodialis Eichl (Menispermaceae) and its alkaloids. Int. Immunopharmacol. 2013, 17, 300-308.
  12. Takhar, P.; Smurthwaite, L.; Coker, H. A.; Fear, D. J.; Banfield, G. K.; Carr, V. A.; Durham, S. R.; Gould, H. J., Allergen drives class switching to IgE in the nasal mucosa in allergic rhinitis. Journal of Immunology. 2005, 174, 5024-5032.

Reviewer 2 Report

The manuscript describes the application of a natural compound isolated from marine resources as a therapeutic agent to potentially treat severe clinical symptoms associated with egg allergy, highlighting its potential to repair the intestinal barrier. The manuscript is easy to follow, although there are some missing words along the manuscript (e.g. line 228 “may be due…” line 279 “may be important”) that should be corrected.

The manuscript seems well designed, with clear presentation of the results and discussion, therefore the manuscript is suitable for publication in Marine Drugs journal after addressing few comments.

It was not clear to me if this compound “viridicatol” was used for the first time as a therapeutical agent for egg allergy in this paper, or if it was already used for other type of food allergy. Please clarify.

Additionally, if the study of egg allergy is the case of allergy model in this manuscript, please detail more information regarding this specific food allergy.

Other comments:

line 41 - correct to "begins"

Line 42 - ovalbumin is the allergen in highest amount but it is considered as a minor allergen. This should be highlighted.

Lines 40-44 - These sentences are well written, but the topic is not clear. Why mentioning the specific case of egg allergy here and them return to the main topic of food allergy.? Maybe rephrase this part of the introduction to make more sense.

Author Response

Response to Reviewer 2 Comments

Point 1: The manuscript describes the application of a natural compound isolated from marine resources as a therapeutic agent to potentially treat severe clinical symptoms associated with egg allergy, highlighting its potential to repair the intestinal barrier. The manuscript is easy to follow, although there are some missing words along the manuscript (e.g. line 228 “may be due…” line 279 “may be important”) that should be corrected. Besides, the whole text were rechecked carefully to avoid the similar missing words in the revised manuscript.

Response 1: Thank you very much for careful reviewing of our manuscript, we are sorry for our poor English. The descriptions “which may due…” and “which may important” were revised as “which may be due…” and “which may be important” in line 248 and line 302 in the revised manuscript respectively.

Point 2: It was not clear to me if this compound “viridicatol” was used for the first time as a therapeutical agent for egg allergy in this paper, or if it was already used for other type of food allergy. Please clarify.

Response 2: Thank you very much for your advice. I’m sorry for our unclarified introduction. At present, the anti-allergic activity of viridicatol has not been reported. The descriptions “As reported, viridicatol effectively inhibited the protein activities and expressions of MMP-2 and -9 to against cancer and tumor [1]. The investigation of biological activity of viridicatol is limited at present.” have been added in line 96-99 in the introduction section in revised manuscript.

Point 3: Additionally, if the study of egg allergy is the case of allergy model in this manuscript, please detail more information regarding this specific food allergy.

Response 3: Thank you very much for your advice. The related descriptions “Eggs are indispensable nutrients in children’s diet pagodas. However, the prevalence of egg allergy in children ranges from 1.2% to 2.0%. The majority allergen of egg were found in egg white, and OVA is the most abundant (58%) one (although not the major allergen), which is claimed to elicit allergic reactions frequently [2]. Thus, the OVA-induced mice allergy model is often used to evaluate the anti-allergic activity of various materials [3, 4] ” were added in line 42-47 in the revised manuscript.

Point 4: line 41 - correct to "begins"

Response 4: Thank you very much for careful reviewing of our manuscript, I’m sorry for this mistake, the "begin" has been modified as "begins" in line 42 in the revised manuscript.

Point 5: Line 42 - ovalbumin is the allergen in highest amount but it is considered as a minor allergen. This should be highlighted.

Response 5: Thank you very much for careful reviewing of our manuscript and we are grateful for your advice. The related description “The majority allergen of egg were found in egg white, and OVA is the most abundant (58%) one (although not the major allergen), which is claimed to elicit allergic reactions frequently [2]” were modified and added in line 44-46 in the revised manuscript.

Point 6: Lines 40-44- These sentences are well written, but the topic is not clear. Why mentioning the specific case of egg allergy here and them return to the main topic of food allergy? Maybe rephrase this part of the introduction to make more sense.

Response 6: Thank you very much for your meaningful advice. We are sorry for the confusing description. The related descriptions were rephrased as “Food allergy is more common to appear in children than in adults, and it often begins in childhood with the influence of genetic predisposition. Eggs are indispensable nutrients in children’s diet pagodas. However, the prevalence of egg allergy in children ranges from 1.2% to 2.0%. The majority allergen of egg were found in egg white, and OVA is the most abundant (58%) one (although not the major allergen), which is claimed to elicit allergic reactions frequently [2]. Thus, the OVA-induced mice allergy model is often used to evaluate the anti-allergic activity of various materials [3, 4]” in line 41-47 in the revised manuscript.

References:

  1. Liang, P.; Zhang, Y. Y.; Yang, P.; Grond, S.; Zhang, Y.; Qian, Z. J., Viridicatol and viridicatin isolated from a shark-gill-derived fungus Penicillium polonicum AP2T1 as MMP-2 and MMP-9 inhibitors in HT1080 cells by MAPKs signaling pathway and docking studies. Med. Chem. Res. 2019, 28, 1039-1048.
  2. Dhanapala, P.; De Silva, C.; Doran, T.; Suphioglu, C., Cracking the egg: An insight into egg hypersensitivity. Molecular. Immunology. 2015, 66, 375-383.
  3. Castillo-Courtade, L.; Han, S.; Lee, S.; Mian, F. M.; Buck, R.; Forsythe, P., Attenuation of food allergy symptoms following treatment with human milk oligosaccharides in a mouse model. Allergy 2015, 70, 1091-1102.
  4. Zhang, T. T.; Hu, Z. Y.; Cheng, Y. W.; Xu, H. X.; Velickovic, T. C.; He, K.; Sun, F.; He, Z. D.; Liu, Z. G.; Wu, X. L., Changes in Allergenicity of Ovalbumin in Vitro and in Vivo on Conjugation with Quercetin. J. Agr. Food Chem. 2020, 68, 4027-4035.

Reviewer 3 Report

The manuscript given for revision is focused on the analysis of the anti-allergic properties of viridicatol using a mice model of albumin-induced food allergy. This compound was able to alleviate the allergy symptoms through suppression of immunoglobulin E, mast cell protease-1, histamine, TNF-α and promoted the production of IL-10.

The manuscript is very well written with logically designed experimental part. The results and discussion fully explain the obtained data.

I have some minor suggestions for corrections listed below:

Page 1, Line 12: Pease correct to: …from 1.2% …

Page 11. Please insert the catalogue numbers of the antibodies in section 4.1.

Please insert it the material and methods section the experimental conditions for NMR, MS and X-ray crystallography.

Author Response

Response to Reviewer 3 Comments

Point 1: Page 1, Line 12: Pease correct to: …from 1.2% …

Response 1: Thank you very much for careful reviewing of our manuscript. I'm sorry for the negligence. The descriptions have been modified in the line 44 in the revised manuscript.

Point 2: Page 11. Please insert the catalogue numbers of the antibodies in section 4.1.

Response 2: Thank you very much for your suggestion. The catalogues numbers and origins of all antibodies have been added in line 358-359 and 363-366 line in the revised manuscript. The descriptions have been added to “Goat anti-mouse IgG1 (ab98693) and IgG2a (ab98698) antibodies were purchased from Abcam (Cambridge, UK)” in the line and “Antibodies for flow cytometry analysis were obtained from BioLegend (San Diego, CA, USA) and BD (New York, NY, USA ). PE anti-mouse CD117(c-Kit) (105807), APC anti-mouse FcεRIα (134316), PE anti-mouse CD19 (115508), Mouse Th/Treg Phenotyping Kit (560767)” in revised manuscript.

Point 3: Please insert it the material and methods section the experimental conditions for NMR, MS and X-ray crystallography.

Response 3: Thank you for kindly suggestion. We are sorry for the simple description in the methods. We are happy to edit the text further based on helpful comments from the reviewers. The material and methods for the NMR, MS, and X-ray crystallography were added and mentioned in the revised Section 2.1. and 2.2., respectively.
